# Enjoyed by Jack but Endured by Jill: An Exploratory Case Study Examining Differences in Adolescent Design Preferences and Perceived Impacts of a Secondary Schoolyard

**DOI:** 10.3390/ijerph20054221

**Published:** 2023-02-27

**Authors:** Gweneth Leigh, Milica Muminovic, Rachel Davey

**Affiliations:** 1Health Research Institute, University of Canberra, Canberra, ACT 2617, Australia; 2Faculty of Arts and Design, University of Canberra, Canberra, ACT 2617, Australia

**Keywords:** schoolyard, design, restorative environment, high school, youth, perception

## Abstract

The school grounds provide students opportunities for respite, relaxation and relief from daily stresses during breaks in the school day. However, it is unclear whether secondary schoolyard designs adequately support the diverse and evolving needs of adolescents, particularly at a time when they are experiencing rapid emotional and physical developmental change. To investigate this, quantitative methods were used to explore differences in perceptions of schoolyard attractiveness and restorative quality based on student gender and year level. A school-wide survey was administered to approximately 284 students in years 7 to 10 at a secondary school in Canberra, Australia. Results indicate significant declines in student perceptions of schoolyard attractiveness and restorative quality. Higher ratings of schoolyard likeability, accessibility, personal connection and restorative quality of ‘being away’ were associated with male students across all year levels. Further work is needed to explore how schoolyard environments can better support the design preferences and well-being needs of older and female students. Such information would help planners, designers and land managers develop schoolyard designs that are more equitable in their benefits to secondary school students of different genders and year levels.

## 1. Introduction

Schoolyard design shapes space in ways that affect how students physically, socially and emotionally engage with one another at school. In a time when peer opportunities for play continue to diminish [1], school grounds provide a place for students to connect with friends and mediate social stresses [2], escape from classroom structures, freely assert their identity and develop a sense of belonging [3]. However, to attain these benefits, schoolyard spaces need to be designed in ways that are inclusive and accessible to the needs of both individual students and the broader school community.

While the design of school buildings continues to evolve in response to environmental and pedagogical needs [4], the pace of change within the design of school grounds has been slow. For decades, most schoolyard design standards have remained focused on the provision of grassed sporting ovals, concrete courtyards and standardized play structures [5,6], with little emphasis on natural diversity [7]. Such spaces are traditionally designed from an adult perspective, guided more by priorities of safety and risk management than around student affordances for physical activity and well-being [3,8,9,10]. As students age, such spaces are increasingly regarded by them as ‘dull and frustrating’ [7].

Time outdoors provides positive health benefits to youth [3,11]. However, the links between schoolyard quality and adolescent health remains understudied, particularly at a time when their mental health issues are growing [12,13]. Students become less active in the schoolyard in secondary school [14]; some have attributed this to low school ground quality [15]. While secondary schoolyards provide students with a daily dose of green space, it is unclear whether its sport-dominated focus is equitable in the affordances provided to students of different genders and ages.

Mounting evidence demonstrates male and female students engage with schoolyard spaces differently. Quantitative tools mapping student physical activity levels reveal male students are more active during recess, while female students are more social and sedentary [2,16,17,18,19]. Several factors contribute towards these differences. In focus group discussions with adolescent girls in Denmark, Pawlowski et al. (2015) found that most did not go into the schoolyard due to ‘a lack of attractive outdoor play facilities’ [20]. In similar discussions at a Melbourne school with Year 6 students, Spark et al. (2019) found girls were excluded from active play spaces by male students [21].

Disparities in schoolyard use also exist based on student age, related in part to the physical and emotional changes associated with adolescence: the developmental needs of students entering Year 7 are quite different to students in Year 10. As students progress in secondary school, there is a shift from engaging in free play to an increased preference for more structured physical activities [5], although research shows this may be influenced by gender. In looking at student behaviors during school breaks, Raney et al. (2023) found that while older male students remained interested in sporting ovals and ball courts, such features did not sustain the interest of older female students [22].

Student perceptions of schoolyard design preferences and its well-being impacts is understudied. Although less documented [23,24,25,26], capturing student voice is recognized as an important attribute in creating effective schoolyard designs [27]. Using perceptions of attractiveness and restoration as indicators of design quality provide an important well-being metric based on user engagement with schoolyard spaces, a missing link among design rating systems that otherwise often focus on sustainability and operational costs [28].

According to Kaplan and Kaplan (1989), aesthetic preference is an indicator of the types of settings in which a person is able to best function [29]. Such places are associated with supporting restoration—the ability to rebuild or renew capabilities that help individuals to function in ways that are physical, cognitive and emotional [30,31]. Their Attention Restoration Theory asserts that time outdoors can help renew focus and alleviate mental fatigue [29]. Its four domains of focus—being away, fascination, extent and compatibility—provide a framework from which to build an understanding of the adequacy of outdoor settings to positively affect user well-being. Research into the attractiveness and perceived restorative quality of open space explores how design features affect the well-being benefits experienced by users [32,33,34,35,36,37]. However, its application to the environments of young people continues to be understudied [23].

This paper applies Kaplan and Kaplan’s restorative environment theory within a traditional school ground setting. It looks to build evidence on the restorative qualities of a secondary schoolyard as perceived and experienced by students, and to understand whether these student views are consistent across the school community. This was explored through the following two questions:Do differences exist in student perceptions of schoolyard attractiveness based on student year level and gender?Do differences exist in student perceptions of schoolyard restorative quality based on student year level and gender?

Schoolyard activities can make an important contribution to the physical, social and emotional development and well-being of students [38]. Schoolyard design quality is formative in these experiences and, as such, can influence the types of health benefits provided to users. Exploring ways to quantify these design impacts based on user experiences provides a metric for identifying where such spaces are performing well, and where additional investment is warranted.

## 2. Materials and Methods

### 2.1. Site Context

The study was structured as an exploratory case study [39] carried out at a public secondary school established in 1985 located in Canberra, Australia. The school consists of approximately 310 students in years 7–10. The school grounds cover approximately 16,000 square meters (51 square meters per child) and consist of a large grass amphitheater, paved ball courts, informal open space with grass and trees, and small shade structures near the canteen, shown in Figure 1. During the school day, students are provided two breaks, recess (thirty minutes) and lunch (forty minutes). Break times are characterized by free play supervised by teachers.

The school was recruited as part of a government initiative called It’s Your Move to co-design with students a multi-use outdoor space in the school grounds. This study was conducted during the conceptual design phase of the schoolyard intervention. Ethics approval was obtained from the Human Research Ethics Committee at the University of Canberra (HREC-6950) and from the ACT government (number RES-2104), followed by permission from the school principal. Participation by students was completely voluntary. A passive consent procedure was conducted through a notice to parents in the school newsletter one week prior to data collection. Students who decided to opt out were instructed to inform their teacher.

### 2.2. Data Collection

The total study population consisted of students aged 11 to 16 in years 7, 8, 9 and 10 (as classified by the Australian educational system). A paper survey was administered to students in each year level as a classroom activity immediately following recess. Collected data were used to investigate student perceptions of schoolyard attractiveness and its perceived restorative impact, with questions outlined further below and in Table 1. Participants provided their demographic information including age, year level and whether they identified as male or female. Survey responses did not require any identifiable student information and remained anonymous. A total of 284 students completed the survey; the teacher of each class collected the data. Each survey was assigned a numerical code during analysis and reporting.

#### 2.2.1. Survey Instrument: Schoolyard Attractiveness

As part of the government schoolyard intervention, a design thinking workshop was facilitated by the ACT government and conducted in August 2020 with a selection of students to identify desired outcomes for the future renewed schoolyard. The top ten student design priorities from this workshop were summarized and incorporated into this study as a series of survey statements (e.g., ‘The design of the schoolyard provides a sense of belonging’ and ‘The design of this schoolyard is accessible by all’). Students undertaking the survey were to consider how each statement applied to their current schoolyard environment and circle the most suitable answer from a 5-point Likert scale, with 1 = Not at all and 5 = Completely true. Students also evaluated the likeability of their schoolyard based on a similar tool used by van Dijk-Wesselius et al. (2018) in studying student appreciation of the school grounds [40]. Schoolyard likeability was rated by students on a 10-point Likert scale from 1 = ‘I don’t like my schoolyard at all’ to 10 = ‘My schoolyard is fantastic, it could not be better’.

#### 2.2.2. Survey Instrument: Schoolyard Restorative Quality

To determine the impact of the school ground design on their individual well-being, students rated the restorative value of the schoolyard. This was based on fifteen items derived from the ‘Perceived Restorative Components Scale for Children’ (PRCS-C II) developed by Bagot, Kuo and Allen (2007) [41]. Adapted from the Perceived Restorativeness Scale developed for adults by Hartig et al. (1997), this validated tool determines the restorative capacity of outdoor settings by measuring the qualities of person–environment transactions within these spaces [42]. Students responded to 15 measures based on the 5 restorative qualities defined by the Attention Restoration Theory [29,43]: Being away–physical, being away–psychological, fascination, extent and compatibility. Students were asked to consider how true each statement was to them and to circle the most suitable answer from a 5-point Likert scale, indicating the extent to which the statement described their experience in the schoolyard with 0 = Not at all and 4 = Completely true.

### 2.3. Data Analysis

Out of the 284 students who participated in the study, valid data were obtained from 252 students aged between 11 and 16 years (M_age_ = 13.59 years, *SD* = 1.16). Students were in year levels between 7 and 10; 52.8 per cent of total participants were male, as indicated in Table 2. Analyses were carried out using IBM SPSS Statistics (Version 27) for Windows.

Descriptive analyses were used to characterize study variables, including means and standard deviations. Unanswered survey questions were assigned a value of zero in the analysis. A principal component analysis (PCA) was conducted to identify broader themes within schoolyard attractiveness questions and to test internal consistencies for schoolyard restorative quality, followed by linear regressions using univariate analysis to examine differences in survey responses based on student year level and gender. This process is outlined further below. Factor loadings after rotation for both categories are provided in Appendix A.

To identify broader themes within the category of Schoolyard Attractiveness, a PCA was conducted with oblique rotation. The Kaiser–Meyer–Olkin measure verified the sampling adequacy for the analysis, KMO = 0.930. An initial analysis was run to obtain eigenvalues for each factor in the data. As not all values were above 0.7, Kaiser’s criterion was deemed too strict and Joliffe’s criterion [44,45] was used instead, retaining all factors with eigenvalues more than 0.7. Two items—schoolyard likeability and accessibility—were removed for separate analysis, as their inclusion negatively impacted analysis reliability due to high eigenvalues but a lack of loading factors. Eight items were retained.

Factor analysis confirmed a bidimensional scale (70.47 per cent explained variance), reducing responses into two factors: ‘Personal connection’ and ‘Design adaptability’. The five design attributes clustered under the factor ‘Personal connection’ focused on those schoolyard impacts that enabled individuals to establish an emotional link with space through feelings of welcome, agency and safety (e.g., ‘The design of this schoolyard provides a sense of belonging’, ‘The design of this schoolyard promotes well-being’, ‘The design of this schoolyard is liberating’). The three attributes defining ‘Design adaptability’ focused on the ability of schoolyard spaces to create flexible environments to accommodate different needs and abilities of students (e.g., ‘The design of this schoolyard is diverse’ and ‘The design of this schoolyard provides age-appropriate play’). Items within each of these two factors were summed and a total schoolyard attractiveness score calculated for each; values could range from 5 to 25 for Personal connection and from 3 to 15 for Design adaptability. The scale for both components showed good reliability, with Cronbach’s alpha, ranging between 0.89 (Personal connection) and 0.75 (Design adaptability).

For the category of schoolyard restorative quality, a PCA was used to test internal consistencies on the 15 items using oblique rotation. The KMO measure verified the sampling adequacy for the analysis, KMO = 0.927; all KMO values for individual items were above the acceptable limit of 0.5 [46]. An initial analysis was run to obtain the eigenvalues for each factor in the data. Four factors had eigenvalues over Jolliffe’s criterion of 0.7, and in combination explained 71 per cent of the variance. Consequently, PRCS-C II measures were reduced from five to four components, with ‘Being away–psychological’ and ‘Being away–physical’ consolidated into one category, ‘Being away’, consisting of six items. Items for each of the three remaining components (fascination, compatibility and extent) align with the factorial structure of PRSC-C II, as identified by Bagot et al. (2007) [41]. These were summed and a total schoolyard perceived restorativeness score calculated. Values could range from 0 to 24 (Being away), 0 to 16 (Fascination), 0 to 8 (Compatibility) and 0 to 8 (Extent). Higher scores indicated higher perceived restorative quality. Reliability was good for all components, with Cronbach’s alpha ranging between 0.73 and 0.88 (Being away: 0.86, Fascination: 0.88, Compatibility: 0.81, Extent: 0.62).

Further investigations were conducted to identify the significance of initial findings of schoolyard attractiveness and restorative quality between sub-groups of students based on gender and year level. The normality of the variables was examined in a preliminary analysis using the Kolmogorov–Smirnov test. Residual plots were also conducted to evaluate the normal distribution of model residuals. As the distribution of responses to both schoolyard attractiveness and restorative quality were significantly non-normal, non-parametric versions of independent sample *t*-tests (Mann–Whitney tests) and analyses of variance (ANOVAs) were used to determine whether significant differences existed. These relationships were further explored through univariate linear regression analyses. The results are presented as unstandardized coefficients (*b*) with 95 per cent confidence intervals (CI). A *p*-value of 0.05 was used to indicate statistical significance.

## 3. Results

Unadjusted means and standard deviations for student responses to schoolyard attractiveness and restorative quality were calculated and are reported in Table 3. Students predominantly scored the schoolyard between the mid-to-lower range of scales. Based on the calculated means of total responses for each category, schoolyard accessibility was the highest rated (M_Accessibility_ = 3.14 on a scale of 1 to 5), and schoolyard fascination was the lowest rated (M_Fascination_ = 5.08 on a scale of 1 to 16).

### 3.1. Significance of Student Year Level and Gender on Perceptions of Schoolyard Attractiveness

In terms of year level, perceptions of schoolyard attractiveness demonstrate significant declines by the time students reach Year 10; results are presented in Table 4. In comparison to Year 10 students, Year 7 students are associated with more positive ratings of schoolyard attractiveness across all categories, with likeability and adaptability both significant at the 1 per cent level; accessibility and personal connection are both significant at the 1 per cent level. These significant differences persist with Year 8 students, predicting more positive ratings of schoolyard likeability at the 5 per cent level (*b* = 1.15, 95% CI 0.20 to 2.11), as well as Year 9 students at the 1 per cent level (*b* = 1.52, 95% CI 0.57 to 2.47) in comparison to Year 10 students.

Significant differences also exist between male and female student perceptions of schoolyard attractiveness. Compared to females, male students predict more positive ratings of schoolyard accessibility at the 1 per cent level, with schoolyard likeability and personal connection at the 5 per cent level.

Further significant relationships are identified when combining the impact of both student gender and year level on perceptions of schoolyard attractiveness. Male students in Year 7 predict higher ratings of schoolyard likeability, accessibility and adaptability at the 5 per cent level when compared to female students in Year 10. This relationship is illustrated in Figure 2.

The differences in the means between male and female students for categories of adaptability, likeability and accessibility were nominal in Year 7, with adaptability at 7.4 per cent (M_MaleAdapt7_ = 9.23 and M_FemaleAdapt7_ = 9.91), likeability at 0.8 per cent (M_MaleLike7_ = 5.14 and M_FemaleLike7_ = 5.18), and accessibility at 1.8 per cent (M_MaleAccess7_ = 3.42 and M_FemaleAccess7_ = 3.36). However, in progressive year levels, the mean schoolyard attractiveness scores of female students decrease at a faster rate than for males. By Year 10, the means of schoolyard attractiveness between male and female students differ by 16.6 per cent for adaptability (M_MaleAdapt10_ = 7.77 and M_FemaleAdapt10_ = 6.48), 23.8 per cent for likeability (M_MaleLike10_ = 4.46 and M_FemaleLike10_ = 3.40), and 25.4 per cent for accessibility (M_MaleAccess10_ = 3.38 and M_FemaleAccess10_ = 2.52).

### 3.2. Significance of Student Year Level and Gender on Perceptions of Schoolyard Restorative Quality

Student perceptions of schoolyard restoration demonstrate significant declines for students between year levels 7 and 10; results are presented in Table 5. In comparison to the mean scores of Year 10 students, students in Year 7 are associated with more positive ratings of schoolyard restorative quality across all categories, with being away and extent both significant at the 1 per cent level and fascination and compatibility both significant at the 5 per cent level.

Significant differences based on student gender were also demonstrated, with male students positively predicting being away at a 5 per cent level of significance. This relationship is further illustrated in Figure 3. In Year 7, there is a 6.3 per cent difference in means between male and female students for the category ‘Being away’ (M_MaleBeingAway7_ = 15, M_FemaleBeingAway7_ = 14.05). By Year 10, this difference expands to 26.3 per cent (M_MaleBeingAway10_ = 13.46, M_FemaleBeingAway10_ = 9.92), as the scores of female students decreased at a faster rate than for males.

Although significant relationships were not identified when combining the impact of student gender and year level on perceptions of restorative quality, there are trends worth noting. In Year 7, male and female students rate restorative qualities similarly except for fascination. Year 7 females scored the schoolyard more than one quarter (26.7 per cent) less fascinating (M_FemaleFasc7_ = 5.27) than male students in the same year (M_MaleFasc7_ = 7.19). This evens out in Year 8, when male fascination scores fall to female Year 8 levels, a decrease of 32.9 per cent (M_MaleFasc8_ = 5.27). However, the restorative scores for Year 8 female students in all other categories continue to decline at rates faster than male students. Between Year 8 and Year 10, female students record drops in schoolyard restorative quality of 17.6 per cent for being away (M_FemaleBeingAway8_ = 12.04 and M_FemaleBeingAway10_ = 9.92), 21.7 per cent in fascination (M_FemaleFasc8_ = 5.11 and M_FemaleFasc10_ = 4.00) and 16.7 per cent drop in extent (M_FemaleExt8_ = 4.80 and M_FemaleExt10_ = 4.00). By comparison, male students between years 8 to 10 demonstrated falls in the same categories of 5.4 per cent (M_MaleBeingAway8_ = 14.23 and M_MaleBeingAway10_ = 13.46), 6.6 per cent (M_MaleFasc8_ = 5.27 and M_MaleFasc10_ = 4.92) and 4 per cent, respectively (M_MaleExt8_ = 5.45 and M_MaleExt10_ = 5.23).

## 4. Discussion

This study examined differences in student perceptions of schoolyard attractiveness and restorative quality based on year level and gender. It aimed to identify how the design of secondary schoolyards is perceived by students and understand whether these views are consistent across the school community. Findings show that students in younger year levels predict more favorable perceptions of schoolyard design and restorative quality compared to students in later year levels. Male students predict more positive perceptions of schoolyard likeability, accessibility and adaptability, as well as the restorative quality of ‘being away’ in contrast to female students. While these results are compatible with related research into student schoolyard behaviors, the findings reveal new insights into student perceptions of schoolyard design.

Previous schoolyard studies have often focused on measuring student use of the school grounds. This study was unique in using student opinion to assess the quality of these experiences. This was achieved through survey measures derived from an adult perspective (schoolyard restorativeness) and student-identified priorities (schoolyard attractiveness). These differences in student schoolyard perceptions may help explain differences in student schoolyard use.

A lack of diversity in secondary schoolyard spaces has been identified as a disincentive to motivate student use [5,47]. For most secondary schools, sporting grounds provide the scaffolding around which schoolyard spaces are frequently organized [5]. The site for this study is no different, with the majority (52.7 per cent) of spaces available to students at break times consisting of open grass, sporting fields and ball courts.

Previous research demonstrates that as students age, traditional schoolyard activities lose appeal [22]. Some of this has been attributed to the rapid social, physical and emotional changes associated with adolescence [48]. It has been proposed that the elevation of structured sports programming within secondary schoolyards fails to accommodate these evolving student needs [49]. Changes to body image, self-esteem and friendship groups can make those with less physical skills more reluctant to participate in sports [50].

During adolescence, the ability for teenagers to retreat from others and escape everyday pressures becomes an important part of their psychological development [51,52,53]. The freedom and independence of outdoor spaces are associated with places of calm and emotional restoration by teenagers [54,55]. Previous studies demonstrate that students have the desire to be active during recess, but find it difficult to achieve within existing designs [56], perceiving there are fewer options available to attract and engage those in higher year levels [57].

The student self-report measures in this study complement the findings of this previous research. The mean of Year 7 student schoolyard attractiveness scores are the highest among year levels; by Year 8, student scores begin a decline from which they do not recover. The positive ratings of Year 7 students could reflect the need of younger students to be physically active [58]; thus, they may appreciate schoolyards more [59].

The decline in student restorative quality scores as they age may also be due to a lack of perceived affordances within the existing schoolyard space. Research by Kaplan and Kaplan (1989) promotes that landscape preferences are indicators of places that best support user needs. The decrease in ratings of schoolyard attractiveness as students age may imply that its lack of appeal creates a lack of apparent restorative benefits to be derived from it.

The disparities in male and female perceptions of schoolyard attractiveness in this study raise the question as to whether schoolyard programming is more naturally aligned with the design preferences of male students. Between years 9 and 10 alone, female students recorded a 31.2 per cent decrease in schoolyard likeability (compared to a drop of 6.5 per cent for males) and a 20.1 per cent decline in schoolyard personal connection (compared to a 5.6 per cent decrease for males). If male students are more attracted across year levels to the sport-dominated design of the study site, then this attraction may also help explain the finding of their more positive perceptions of ‘being away’ during school breaks compared to female students.

The differences in male and female attitudes to play are well documented. Boys have traditionally been more physically active and reliant on sporting fields across year levels [47,60,61,62]. By contrast, female students have been documented as relying less on—and even avoiding—areas of turf and asphalt [61,63,64,65,66,67,68], with play more focused on sedentary games within small groups [22]. Their preferences are more aligned with non-competitive play activities that provide greater choice in activities over traditional sports, such as obstacle courses, trampolines, dancing and gymnastics [22,68,69,70]. Such findings suggest that the large land areas devoted to ball sports might actually come at the cost of minimizing female physical activity choices [22].

In contrast to boys, research shows girls spend much less time outdoors [71], while also demonstrating worsening mental health, particularly in terms of psychological distress and life satisfaction [72]. These disparities often extend into adulthood: compared to men, women are three times more likely to experience common mental health problems, particularly in countries considered gender equal [72]. Given the known health benefits of time outdoors, the findings of this study raise whether female students have ‘more to gain’ [73] from improvements to schoolyard spaces than male students.

Research demonstrates that the outdoor preferences of adolescents can be quite variable across ages [55]. Versatile schoolyard spaces that provide diverse, multi-sensory experiences have been shown to provide a greater choice of activities [10] and be more effective in reducing student stress and promoting creativity [38]. Considering the significant differences in student perceptions of schoolyard attractiveness and restorative quality found in this study, increasing the diversity of available schoolyard settings might be one way of increasing the benefits attained by students during outdoor breaks in the school day.

While this study makes contributions in the knowledge around adolescent design preferences and perceived impacts of schoolyards, it also has limitations. As an exploratory study, its limited scope and small sample size may constrain inferences from produced regression models. Given the different types of environments available in the schoolyard (open/paved/sheltered), measuring student perceptions within each would have provided context on which areas best supported user needs. However, due to practicalities of availability of students, time constraints and staff resourcing, this was not possible. It would be beneficial to expand and diversify the sample to a broader subset of schools of different sizes and locations to compare whether findings from this study hold to larger populations. Better monitoring of recess conditions during data collection would help limit confounding factors that may affect results, such as the impact of food during break times, peer effects and varying amounts of time spent by students outdoors prior to undertaking the survey. Given the quantitative nature of the study, results focus on broader patterns of age and gender differences. Qualitative studies that provide students the opportunity to share their perspectives and experiences would complement these findings and explore possible determinants of student schoolyard perceptions and preferences.

## 5. Conclusions

Adolescence is a critical period of social, emotional and physical development. The findings from this study are important for questioning whether existing schoolyard design standards positively support the changing needs of secondary students. Previous studies have addressed the adequacy of schoolyard design by measuring student physical activity levels and observing behavioral patterns. This study is unique by focusing on how students, as users, rate the quality of their schoolyard experiences.

Study findings reveal new insights into differences in schoolyard perceptions between younger and older students, as well as between male and female schoolyard users. Compared to Year 7 students, Year 10 students demonstrated significant declines in ratings across all categories of schoolyard attractiveness and restorative capacity. Female students are more negative than male students in their perceptions of schoolyard accessibility, likeability and personal connection. Male students more positively view the restorative design capacity of the schoolyard to support feelings of ‘being away’. These results suggest that schoolyard standards need revising to provide optimal environments that fulfill the design preferences of females and older students.

The schoolyard is a social landscape. Its design can help students build peer interactions, feel a sense of belonging and find respite during long days of classes. However, to help students thrive, its design needs to consider how issues of materiality, space, form and programming responds to the diverse needs of the broader student body. While the benefits of spending time outside are well known, less understood are the impacts of how its design quality affects user experiences. Exploring new metrics that account for student voice in assessing schoolyard quality can help validate where designs are working well, while providing the tools for calling it out when it is not.

## Figures and Tables

**Figure 1 ijerph-20-04221-f001:**
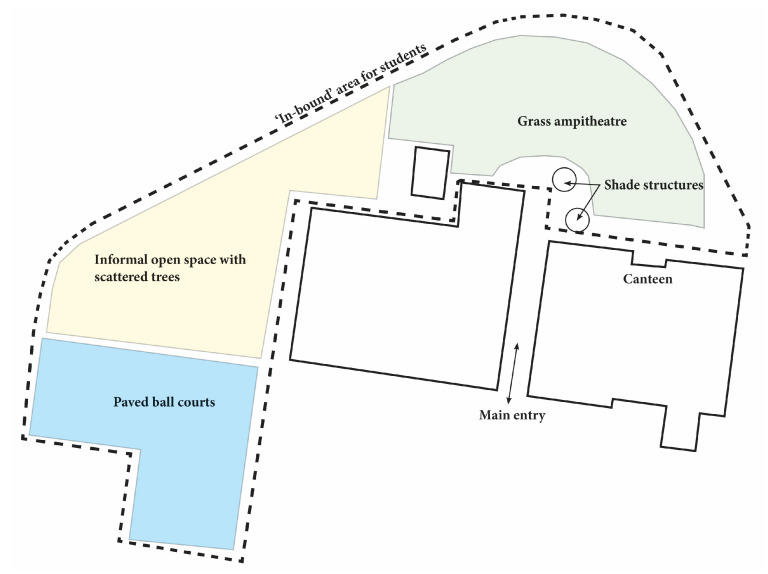
Aerial map of the study site. Dashed line indicates the schoolyard areas where students are allowed to go during recess and lunch breaks.

**Figure 2 ijerph-20-04221-f002:**
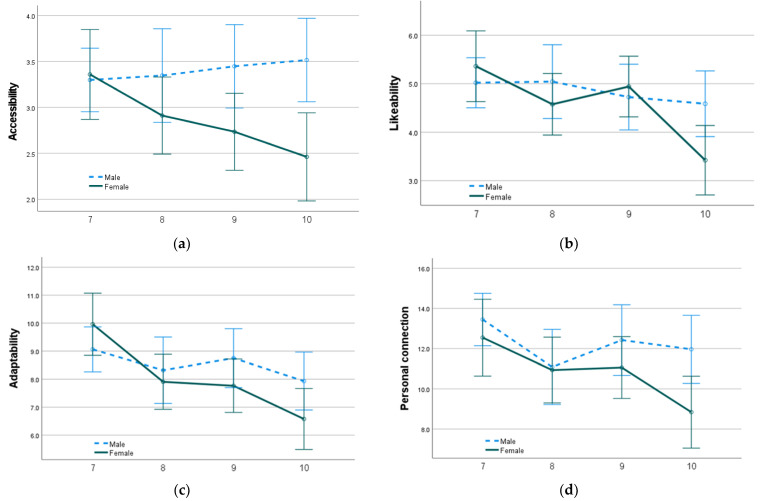
Mean scores of male and female student schoolyard perceptions in each year level (7, 8, 9 and 10) for (**a**) Accessibility, (**b**) Likeability, (**c**) Adaptability and (**d**) Personal connection. Higher scores indicate a more positive opinion of schoolyard attractiveness. Error bars represent 95% CI.

**Figure 3 ijerph-20-04221-f003:**
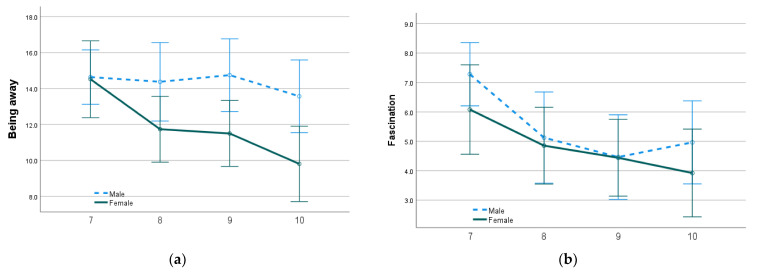
Mean scores of male and female student schoolyard perceptions in each year level (7, 8, 9 and 10) for (**a**) Being away, (**b**) Fascination, (**c**) Compatibility and (**d**) Extent. Higher scores indicate a more positive opinion of schoolyard restorative quality. Error bars represent 95% CI.

**Table 1 ijerph-20-04221-t001:** Student survey questions.

Category	Survey Questions
Personal connection	The design of this schoolyard encouraged me to be adventurous
	The design of this schoolyard provides a sense of belonging
	The design of this schoolyard is engaging
	The design of this schoolyard promotes well-being
	The design of this schoolyard is liberating
Design adaptability	The design of this schoolyard is diverse
	The design of this schoolyard provides age-appropriate play
	The design of this schoolyard promotes physical activity
Likeability	Please rate your opinion of the schoolyard with one being the worst and ten being the best
Accessibility	The design of this schoolyard is accessible by all
Fascination	There are lots of interesting places in the school ground
	There are lots of things to discover in the school ground
	There are lots of interesting things to look at in the school ground
	There are many things in the school ground that I find fascinating
Being away	When I am in the school ground, it feels as though I am in different surroundings than when I am in the classroom
	When I am in the school ground, it feels as though I am in a different place than in the classroom
	When I am in the school ground, I do different things than in the classroom
	When I am in the school ground, I feel free from all the things teachers want me to do
	When I am in the school ground, I feel free from schoolwork and class time
	When I am in the school ground, I am away from things I must do
Compatibility	The things I like to do can be done in the school ground
	The things I want to do can be done in the school ground
Extent	I can do many different things in one part of the school ground
	I can think of all the different areas of the school ground as like lots of little school grounds joined together
	I do different things in different areas of the school ground

**Table 2 ijerph-20-04221-t002:** Descriptive sample characteristics (*N* = 252).

Year Level	Males		Females		Total Students	% Total Students
	Total	% Year	Total	% Year		
7	50	66.7%	25	33.3%	75	29.8%
8	25	42.4%	34	57.6%	59	23.4%
9	29	46.0%	34	54.0%	63	25.0%
10	29	52.7%	26	47.3%	55	21.8%
Total	133		119		252	100.0%

**Table 3 ijerph-20-04221-t003:** Unadjusted means and standard deviations of student survey responses.

Survey Measure	Year 7		Year 8		Year 9		Year 10	
	Males	Females	Males	Females	Males	Females	Males	Females
Attractiveness								
Likeability	5.14 ± 2.07	5.18 ± 1.99	5.00 ± 1.35	4.68 ± 1.79	4.77 ± 1.82	4.94 ± 2.03	4.46 ± 1.50	3.40 ± 1.87
Accessibility	3.42 ± 1.37	3.36 ± 1.40	3.36 ± 0.95	2.82 ± 1.21	3.54 ± 1.14	2.74 ± 1.29	3.38 ± 1.30	2.52 ± 1.16
Adaptability	9.23 ± 3.22	9.91 ± 2.79	8.31 ± 2.10	7.89 ± 3.18	8.77 ± 2.66	7.76 ± 3.06	7.77 ± 2.70	6.48 ± 2.65
Personal Connection	13.63 ± 5.01	12.55 ± 4.94	10.86 ± 4.02	11.36 ± 4.73	12.42 ± 4.57	11.06 ± 4.53	11.73 ± 4.50	8.84 ± 3.47
Restorative quality								
Being away	15.00 ± 6.01	14.05 ± 5.95	14.23 ± 4.45	12.04 ± 6.22	14.62 ± 5.04	11.5 ± 5.00	13.46 ± 5.41	9.92 ± 5.57
Fascination	7.19 ± 4.77	5.27 ± 3.61	5.27 ± 3.45	5.11 ± 3.56	4.46 ± 3.75	4.44 ± 3.96	4.92 ± 3.59	4.00 ± 3.38
Compatibility	3.74 ± 2.56	4.00 ± 2.02	3.55 ± 1.95	2.89 ± 1.91	3.42 ± 2.32	3.38 ± 2.34	3.42 ± 2.23	2.72 ± 2.13
(Extent)	6.00 ± 3.16	5.55 ± 3.00	5.45 ± 2.41	4.80 ± 2.66	5.27 ± 3.04	4.97 ± 2.81	5.23 ± 2.89	4.00 ± 3.11

**Table 4 ijerph-20-04221-t004:** Unstandardized regression coefficient (*b)* for schoolyard attractiveness. Means indicated with an asterisk indicate significant differences with * *p* ≤ 0.05, ** *p* ≤ 0.01 and *** *p* ≤ 0.001. CI = Confidence interval. Females and Year 10 are the reference group. Bold indicates the relationship is significant.

	Likeability		Accessibility		Design Adaptability		Personal Connection	
	*b*	95% CI	*b*	95% CI	*b*	95% CI	*b*	95% CI
Gender (male)	**1.16 ***	**0.18 to 2.15**	**1.06 ****	**0.40 to 1.72**	1.35	−0.15 to 2.86	**3.12 ***	**0.66 to 5.59**
Year 7 (total)	**1.94 *****	**0.92 to 2.96**	**0.9 ****	**0.21 to 1.58**	**3.38 *****	**1.83 to 4.94**	**3.71 ****	**1.09 to 6.33**
Year 7 males	**−1.5 ***	**−2.83 to −0.17**	**−1.12 ***	**−2.01 to −0.22**	**−2.25 ***	**−4.29 to −0.22**	−2.22	−5.61 to 1.16
Year 8 (total)	**1.15 ***	**0.20 to 2.11**	0.45	−0.19 to 1.09	1.33	-0.14 to 2.80	2.09	−0.33 to 4.52
Year 8 males	−0.7	−2.10 to 0.70	−0.62	−1.55 to 0.32	−0.94	-3.09 to 1.21	−2.97	−6.47 to 0.53
Year 9 (total)	**1.52 ****	**0.57 to 2.47**	0.27	−0.36 to 0.91	1.19	−0.26 to 2.64	2.22	−0.14 to 4.58
Year 9 males	**−1.38 ***	**−2.73 to −0.03**	−0.34	−1.25 to 0.56	−0.37	−2.43 to 1.70	−1.76	−5.16 to 1.64
R^2^	**0.072 ***		**0.074 ****		**0.095 *****		**0.082 ****	

**Table 5 ijerph-20-04221-t005:** Unstandardized regression coefficient (*b)* for schoolyard restoration. Means indicated with an asterisk indicate significant differences with * *p* ≤ 0.05, ** *p* ≤ 0.01 and *** *p* ≤ 0.001. CI = Confidence interval. Females and Year 10 are the reference group. Bold indicates the relationship is significant.

	Being Away		Fascination		Compatibility		Extent	
	*b*	95% CI	*b*	95% CI	*b*	95% CI	*b*	95% CI
Gender (males)	**3.76 ***	**0.85 to 6.68**	1.04	−1.01 to 3.10	0.74	−0.44 to 1.92	1.23	−0.30 to 2.79
Year 7 (total)	**4.71 ****	**1.72 to 7.71**	**2.16 ***	**0.03 to 4.29**	**1.43 ***	**0.21 to 2.64**	**2.12 ****	**0.52 to 3.72**
Year 7 males	−3.64	−7.56 to 0.27	0.16	−2.62 to 2.93	−1.14	−2.73 to 0.46	−1.23	−3.31 to 0.86
Year 8 (total)	1.93	−0.86 to 4.71	0.93	−1.05 to 2.91	0.09	−1.04 to 1.22	0.58	−0.91 to 2.07
Year 8 males	−1.12	−5.20 to 2.95	−0.77	−3.66 to 2.12	0.3	−1.35 to 1.96	−0.58	−2.75 to 1.59
Year 9 (total)	1.69	−1.09 to 4.48	0.52	−1.46 to 2.50	0.77	−0.36 to 1.90	1.01	−0.48 to 2.50
Year 9 males	−0.51	−4.51 to 3.48	−1.02	−3.84 to 1.81	−0.64	−2.25 to 0.97	−0.77	−2.88 to 1.35
R^2^	**0.092** ***		**0.081** **		0.04		0.054	

## Data Availability

Not applicable.

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
