# Peer review of "Enjoyed by Jack but Endured by Jill: An Exploratory Case Study Examining Differences in Adolescent Design Preferences and Perceived Impacts of a Secondary Schoolyard"

_ijerph, 2023, doi:10.3390/ijerph20054221_

Round 1

Reviewer 1 Report

This study contributed to the knowledge of adolescent design preferences and the perceived impacts of schoolyards. The scope and methods of the research are clear, and the limitations of the results are accurately recognized.

The authors should better emphasize the uniqueness of this study relative to previous studies in the conclusions.

 Minor Correction

The title of Table. 4, “**p≤..01 and ***p≤..001” should be “**p≤.01 and ***p≤.001”.

Please unify the decimal point notation on the axis scales of graphs, for example, 3.00 or 3.

Comments This study contributed to the knowledge of adolescent design preferences and the perceived impacts of schoolyards. The scope and methods of the research are clear, and the limitations of the results are accurately recognized. Line number 321-323 "While these results are compatible with related research into student schoolyard behaviors, the findings reveal new insights into student perceptions of schoolyard design." The authors should better emphasize the uniqueness of this study relative to previous studies in the Conclusions. The authors explained the new findings in some parts of the Discussion, but please briefly summarize them in the Conclusions section. The current Conclusions seem indistinguishable from the findings of previous studies, and it is difficult to tell what has been added to the subject area by this study.

Reviewer 2 Report

Suggestions in the file attached
